# Zn^2+^-Dependent Nuclease Is Involved in Nuclear Degradation during the Programmed Cell Death of Secretory Cavity Formation in *Citrus grandis* ‘Tomentosa’ Fruits

**DOI:** 10.3390/cells10113222

**Published:** 2021-11-18

**Authors:** Minjian Liang, Mei Bai, Hong Wu

**Affiliations:** State Key Laboratory for Conservation and Utilization of Subtropical Agro-Bioresources, Laboratory for Lingnan Modern Agriculture, College of Life Sciences, South China Agricultural University, Guangzhou 510642, China; yfdfh96@163.com

**Keywords:** secretory cavity, PCD, CgENDO1, Zn^2+^ ions, nuclear DNA degradation

## Abstract

Zn^2+^- and Ca^2+^-dependent nucleases exhibit activity toward dsDNA in the four classes of cation-dependent nucleases in plants. Programmed cell death (PCD) is involved in the degradation of cells during schizolysigenous secretory cavity formation in *Citrus* fruits. Recently, the Ca^2+^-dependent DNase CgCAN was proven to play a key role in nuclear DNA degradation during the PCD of secretory cavity formation in *Citrus grandis* ‘Tomentosa’ fruits. However, whether Zn^2+^-dependent nuclease plays a role in the PCD of secretory cells remains poorly understood. Here, we identified a Zn^2+^-dependent nuclease gene, *CgENDO1*, from *Citrus* *grandis* ‘Tomentosa’, the function of which was studied using Zn^2+^ ions cytochemical localization, DNase activity assays, in situ hybridization, and protein immunolocalization. The full-length cDNA of *CgENDO1* contains an open reading frame of 906 bp that encodes a protein 301 amino acids in length with a S1/P1-like functional domain. CgENDO1 degrades linear double-stranded DNA at acidic and neutral pH. *CgENDO1* is mainly expressed in the late stage of nuclear degradation of secretory cells. Further spatiotemporal expression patterns of CgENDO1 showed that CgENDO1 is initially located on the endoplasmic reticulum and then moves into intracellular vesicles and nuclei. During the late stage of nuclear degradation, it was concentrated in the area of nuclear degradation involved in nuclear DNA degradation. Our results suggest that the Zn^2+^-dependent nuclease CgENDO1 plays a direct role in the late degradation stage of the nuclear DNA in the PCD of secretory cavity cells of *Citrus grandis* ‘Tomentosa’ fruits.

## 1. Introduction

Programmed cell death (PCD) is an intracellular program for death, which indicates that a cell executes a specific physiological process under the control of its own genes. It plays a very important role in the natural growth, development, and aging of plants and in reactions to pathogens [1,2,3]. Cells executing PCD exhibit a series of morphological and biochemical changes, among which the most typical morphological feature is the degradation of the nucleus, including chromatin condensation, DNA fragmentation, and nuclear membrane degradation [2,4,5].

In animal and plant PCD cells, the degradation of the nucleus can be roughly divided into three stages. The first is the occurrence and increase of chromatin condensation, followed by DNA fragmentation, and finally the complete degradation of the nucleus [6,7]. In animal cell apoptosis, the mechanism of nuclear degradation mainly requires caspase-activated DNase (CAD) nuclease and DNase γ. Caspase-3 activates CAD nuclease and DNase γ to degrade the DNA between the nucleosomes by splicing the inhibitory protein of CAD nuclease (ICAD) [8,9]. However, no nuclease similar to animal CAD has been found to be involved in plant PCD. On the other hand, metacaspases and paracaspases are ancestors of caspases. They have the caspase-hemoglobinase fold, but they show distinct substrate specificity and activation mechanisms [10]. In plants, metacaspases cooperate with autophagy to regulate cell aging, immune responses, terminal differentiation, and postmortem cell clearance [10]. Interestingly, caspase-like proteases were reported to execute PCD in plant cells [11].

Recently, we found that CgPBA1, a proteasome with caspase-3-like activity in *Citrus grandis* ‘Tomentosa’, may be involved in the degradation of cell nuclei in secretory cavity epithelial cells [12]. The mechanism of nuclear DNA and RNA degradation in plant PCD mainly focuses on the role of ion-dependent nucleases. There are four divalent cation-dependent nuclease types in plants, among which only Ca^2+^- and Zn^2+^-dependent nucleases are involved in double-stranded DNA degradation [13]. Ca^2+^-dependent nucleases effectively act on double-stranded DNA (dsDNA) under neutral and optimal pH conditions [14], while Zn^2+^-dependent nucleases mainly act on single-stranded DNA (ssDNA) and RNA under acidic and optimal pH conditions [15]. However, no protein similar to Zn^2+^-dependent nuclease in the late PCD stage of plant PCD has been found in animal PCD [8].

Zinc is a necessary component of the structure of certain proteins in animal and plant cells and is necessary to activate zymogen in enzymatic reactions [16,17]. Zinc generally regulates cell activities in the form of zinc divalent cations (Zn^2+^ ions) through transient changes in concentration [18]. Therefore, as a major intracellular regulatory ion, Zn^2+^ ions can participate in a series of biological redox reactions in the body and form a special zinc protein grid in the cell, which works together with all of the other biological action pathways to complete cell metabolism, and even affect the cell fate [19].

The amino acid sequence of Zn^2+^-dependent nucleases in most plants is highly similar to S1 nuclease in *Aspergillus oryzae* and P1 nuclease in *Penicillium citrinum* [20], which is referred to as S1/P1-like nuclease in plants. S1/P1-like nucleases have high nuclease activity under acidic pH conditions and can specifically cleave ssDNA, RNA, and mismatched dsDNA [21]. However, they have certain differences in their ability to digest dsDNA, ssDNA, and RNA. The activity of S1/P1-like nuclease does not depend on the cDNA expression level of the enzyme, but on its ion concentration, pH, and substrate concentration. The stability of its C-terminus may be an important factor related to the activity of S1/P1-like nucleases in plants [21,22,23,24,25,26,27,28].

Studies have shown that there are five Zn^2+^-dependent S1/P1-like nucleases in *Arabidopsis thaliana*, namely, ENDO1 (BFN1, At1G11190), ENDO2 (At1G68290), ENDO3 (At4G21590), ENDO4 (At4G21585), and ENDO5 (At4G21600) [23]. They are involved in the differentiation, development, and maturation of flowers and fruits [24,26,29], the degradation of the endosperm during seed development [30], and early nuclear degradation during unhealthy and aging cells PCD, which is also associated with cytoplasmic shrinkage and chloroplast polymerization or degradation [31]. Zn^2+^-dependent nuclease *Zinnia* endonuclease 1 (ZEN1) plays a role in nuclear degradation during tracheary element (TE) differentiation [13]. The Zn^2+^-dependent nuclease *barley* endonuclease 1 (BEN1) plays a role in the nuclear DNA degradation of endosperm cells during seed germination and is expressed together with Bunc1 and Bunc2 in the formation of barley microgametocytes. In addition, it is involved in the process of PCD in developing anthers [32,33,34,35]. Moreover, an acidic Zn^2+^-dependent nuclease was found in epidermal and parenchyma cells PCD during wheat grain germination [36].

The secretory cavity is a common structural feature in *Citrus* fruits, which is an important place for the synthesis and accumulation of medicinal active substances [37]. Fruit secretory cavities are formed schizolysigenously under the action of pectinase and cellulase [38,39]. Among them, PCD is involved in the degradation of secretory cavity cells during the formation of secretory cavities of *C. grandis* ‘Tomentosa’ and *C. sinensis* (L.) Osbeck fruits [40,41].

Multiple nucleases were required for nuclear degradation in PCD cells. Sugiyama et al. suggested that in plant cells with PCD, Ca^2+^- or Mg^2+^-dependent neutral nucleases were first targeted to the nucleus, and then Zn^2+^-dependent acid nucleases were activated after vacuolar membrane cleavage [15]. As signaling molecules, calcium ions are involved in plant PCD in a variety of ways, including developmental PCD regulated by internal factors and environmental PCD induced by external stimuli [42]. Calcium signaling can affect the proteins responsible for nuclear transport to mediate PCD in plant stress resistance [43]. Calcium ions regulate nuclear DNA degradation in PCD [42]. Recently, it was found that the Ca^2+^-dependent nuclease CgCAN is directly involved in the degradation of nuclear DNA in the PCD of secretory cavity cells in *C. grandis* ‘Tomentosa’ fruits [14]. Unfortunately, it has not been determined whether Zn^2+^-dependent nucleases are present in *C. grandis* ‘Tomentosa’.

The aim of this study was to determine how Zn^2+^-dependent nucleases are involved in the PCD of secretory cavity cells during the development of the secretory cavity. We cloned the Zn^2+^-dependent nuclease gene *CgENDO1*, which is involved in the development of secretory cavity cells PCD in *C. grandis* ‘Tomentosa’ fruits. Its ORF contains 906 bp and it encodes 301 amino acids. In addition, it has the zinc ion-dependent nuclease activity of CgENDO1. Moreover, we demonstrated the intracellular free Zn^2+^ ions subcellular localization for the first time. Furthermore, we used in situ hybridization for histological localization of CgENDO1 and immunocytochemical localization for subcellular localization of Zn^2+^-dependent nucleases. The results indicate that Zn^2+^ ions activate the Zn^2+^-dependent nuclease CgENDO1 for the involvement in the degradation of nuclear DNA in the secretory cavity cell PCD of *Citrus grandis* ‘Tomentosa’ fruits.

## 2. Materials and Methods

### 2.1. Plant Materials and Sampling

The flowers and fruits of a 15-year-old *Citrus grandis* ‘Tomentosa’ tree were collected from the farm of South China Agricultural University, Guangzhou, China (N 23°09′47.84′′, E 113°22′12.58′′). The ovary wall of the flowers and fruit exocarp at different developmental stages were collected and labeled H1-H12 (Appendix A).

### 2.2. Quantitative Real-Time PCR Analysis of *CgENDO1*

Total RNA was extracted using the Column Plant RNAout 2.0 kit (TIANDZ). The RNA was detected by an ultramicroscopic spectrophotometer Nanodrop™ One (Thermo, Waltham, MA, USA). The PrimeScriptTM RT Reagent Kit with gDNA Eraser (Takara, Beijing, China) was used to synthesize cDNA. The primers for the target gene and internal reference gene are shown in Appendix A. The iTaq^TM^ Universal SYBR Green Supermix kit (BioRad, Hercules, CA, USA) was used for real-time quantitative PCR (RT-PCR). Each experiment had three sample replicates and three technical replicates, and the relative expression was calculated using 2^−∆∆Ct^. The SPSS 21.0 software was used for the T-tests and the analysis of variance.

### 2.3. Cloning and Sequence Analysis of *CgENDO1*

The method of total RNA extraction and reverse transcription of the first strand cDNA was the same as 2.2. The gene sequences of *Arabidopsis thaliana ENDO1* (*BFN1*, AT1G11190) were obtained from the NCBI database (https://www.ncbi.nlm.nih.gov/; accessed on 30 September 2021), and the *Citrus sinensis* genome database in BLAST was used to get the sequence of *CsENDO1* (LOC102622496). We used the Primer Premier 5 software to design the primers (shown in Appendix A). The 906 bp cDNA sequence was cloned using TaKaRa Taq Version 2.0 plus dye (TaKaRa, Beijing, China). The amino acid sequence was translated by the DNAMAN8.0 software (LynnonBiosoft, CA, USA) and named CgENDO1. Then, the NCBI protein in the conservative domain database (https://www.ncbi.nlm.nih.gov/cdd; accessed on 30 September 2021) was used for the CgENDO1 protein function structure domain and active site prediction, and in the NCBI BLAST, the homologous proteins of the other species were removed and the DNAMAN8.0 blast function domains were used.

### 2.4. In Vitro Expression of *CgENDO1* in Escherichia coli and Enzyme Activity Assay

The complete ORF of *CgENDO1* was cloned into the pGEX-4T1 vector using the ClonExpress Ultra One Step Cloning Kit (Vazyme, Nanjing, China), and the recombinant primers used are shown in Appendix A. Then, we expressed it in *E. coli* Rosetta (DE3). The fusion protein GST-CgENDO1 was expressed for 5 h at 37 °C under the induction of 0.5 mM isopropyl-b-d-1-thiogalactopyranoside (IPTG). A GST-Sefinose (TM) Kit (Sangon Biotech, Shanghai, China) was used to purify the recombinant protein CgENDO1 with GST. One microgram of purified protein was mixed with the same amount of different types of DNA (*Citrus grandis* ‘Tomentosa’ genomic DNA and T65 genomic DNA, BD circular plasmid and linear BD plasmid DNA by EcoRI restriction enzymes). Different concentrations of Zn^2+^ ions (with or without 2 M, 1 M, 500 mM, 100 mM, 10 mM, 5 mM or 1 mM ZnCl_2_) were added to the reaction buffer (pH 5.5 and 8.0). The solution was digested at 37 °C for 1 h and detected by 1% agarose gel electrophoresis. The negative control was performed under the same reaction conditions and buffer system without protein purpose or the plus GST tag reaction, and the positive control was treated with DNaseI (Sangon Biotech, Shanghai, China) to degrade the same amount of DNA substrate.

### 2.5. Morphological and Developmental Observation and Immunocytochemical Localization

Different growth stages of the ovary wall of *Citrus grandis* ‘Tomentosa’ flowers and exocarps were divided into small blocks (0.5 × 0.5 × 0.5 mm) and then fixed in 4% paraformaldehyde and 0.5% glutaraldehyde (0.1 M PBS, pH 7.2). After a series of alcohol dehydrations, the samples were embedded in Epon 812 epoxy resin. The samples were cut into 1 μm thick sections using a Leica RM2155 thin slicer (Leica, Wetzlar Hesse-Darmstadt, Germany), stained with 0.01% toluidine blue, and observed under a microscope by Leica DMLB (Leica, Wetzlar Hesse-Darmstadt, Germany).

Ultrathin sections were cut 70–80 nm thick with a Leica UC6 (Leica, Wetzlar Hesse-Darmstadt, Germany). A nickel grid containing the sections was washed in 8% sodium periodate for 4 min at room temperature, three times with ddH_2_O, PBST twice, and then it was blocked with 1% bovine serum albumin (BSA) for 50 min. The nickel grid was washed with PBST twice and floated on PBST containing anti-CgENDO1 specific polyclonal antibodies (primary antibodies, 1:30; *v*/*v*) for 3 h at 37 °C and then washed with PBST three times. The nickel grid was floated on PBST containing colloidal gold antibodies (secondary antibodies, 1:50; *v*/*v*, 10 nm gold particles, Sigma-Aldrich, Shanghai, China) and incubated for 1 h at 37 °C. The nickel grid was washed with PBST three times and washed with ddH_2_O three times. The nickel grid was stained with uranyl acetate and lead citrate. Control samples were treated in a similar manner. In control A, the primary antibody was replaced with preimmunization serum. In control B, the primary antibody was replaced with PBS. The sections were examined and photographed using a TALOS L120C transmission electron microscope (Thermo, Waltham, MA, USA).

### 2.6. *CgENDO1* Expression Analysis by In Situ Hybridization

In different growth stages, the ovary wall of flowers and exocarp were cut into small pieces (5 × 5 × 5 mm). The samples were fixed in 4% paraformaldehyde and embedded in paraffin. They were then cut into 5 μm sections. In situ hybridization was performed according to the protocols described by Bai et al. [14]. A 213 bp fragment from position 285 to 497 bp of CgENDO1 was used to generate a probe. The labeled signal was observed and photographed with a Leica DM6B light microscope (Leica DM6B, Leica, Wetzlar, Germany).

### 2.7. Zn^2+^ Ions Subcellular Localization

Heavy metals were transformed into silver sulfide for subcellular localization using the silver amplification method, as described by Pihl E. [44]. Intracellular free Zn^2+^ ions were converted to Ag_2_S precipitation by a substitution reaction. Under transmission electron microscopy (TEM), Ag_2_S precipitation is displayed as a black solid precipitation. The Ag_2_S black particle distribution location shows the location of the original free Zn^2+^ ions inside and outside the cells [44]. The basic principle is:H_2_S + Zn^2+^ **→** ZnS + 2H^+^
ZnS + 2Ag^+^ **→** Ag_2_S**↓**+ Zn^2+^

The specific operation we improved was as follows: The ovary wall and the exocarp of *Citrus grandis* ‘Tomentosa’ at different developmental stages were taken and divided into 0.5 × 0.5 × 1 mm pieces, which were fixed in 5% glutaraldehyde fixation solution (prepared with H_2_S saturated solution) at 4 °C for 24 h. The sample was washed with PBS (pH 7.2) five times for 20 min each time. The samples were incubated with an incubation solution (30% arabic acid, 0.015 M hydroquinone, 0.022 M citric acid, 0.293 M sucrose, pH 3.9–4.0, and a 10% silver nitrate solution was added only before use) at 37 °C for 1 h. The samples were washed with d_2_H_2_O five times for 20 min each and fixed overnight at 4 °C with 1% osmium acid. The samples were washed with PBS (pH 7.2) five times for 20 min each, dehydrated with an ethanol series, embedded in excessive propylene oxide, embedded in Epon 812 resin, and polymerized at 40 °C for 24 h and 60 °C for 24 h. The sections were cut at 90 nm with a Leica UC6 (Leica, Wetzlar Hesse-Darmstadt, Germany) ultrafine slicer. Double staining was applied with uranium acetate and lead citrate. Control samples were treated in the same way.

To prove that the black particles we observed under TEM contain Ag, which represents the location of free Zn^2+^ ions in cells, a Talos F200S field emission transmission electron microscope (Thermo, Waltham, MA, USA) for energy dispersive spectroscopy (EDS) was applied. Under TEM, black particles were observed for the contained energy spectrum analysis and identification of the elements. We found that the black precipitate particles in both the nucleus and the cell wall, all contained Ag elements (Appendix A, arrowhead). As described in the experimental method, we replaced the intracellular Zn^2+^ ions with Ag_2_S precipitates, which proved that the black particle precipitates inside and outside the cells were indeed Ag elements, which represented the location of Zn^2+^ ions.

## 3. Results

### 3.1. Zn^2+^-Dependent Nuclease CgENDO1 Degrades DNA

The development process of the secretory cavity of *C. grandis* ‘Tomentosa’ fruits can be divided into six stages: The early initial cell stage, the middle initial cell stage, the late initial cell stage, the lumen-forming stage, the lumen-expanding stage, and the mature stage [14]. Based on the size of the ovaries and young fruits, the samples were divided into 12 different growth stages, and the statistics of the proportion of the development process of the secretory cavity in different sizes of the samples were obtained (Appendix A). We used qRT-PCR of *Citrus grandis* ‘Tomentosa’ fruit exocarp in different growth periods to measure the relative quantitative transcription levels of Zn^2+^-dependent nucleases. We found that *CgENDO1* had specifically high expression levels in the late initial cell stage and the lumen-forming stage. Moreover, the expression level of *CgENDO1* was very low in endocarp without secretory cavities (Figure 1A).

We cloned Zn^2+^-dependent nuclease *CgENDO1* from *C. grandis* ‘Tomentosa’. The full-length cDNA ORF contains 906 bp, encoding a protein with 301 amino acids (Figure 1Ba). The amino acid sequence of CgENDO1 is 99% similar to *C. sinensis* (L.) Osebeck (Figure 1Bb). According to the prediction results of the protein structure in NCBI, the CgENDO1 sequence in *C. grandis* ‘Tomentosa’ contains a typical S1/P1 nuclease domain belonging to the S1/P1 nuclease superfamily, with nine active sites (Trp-1, His-6, Asp-45, His-60, His-116, Asp-120, His-126, His-149, and Asp-153) (Figure 1Bc).

To further verify the function of CgENDO1, we detected the digestion activity of the fusion protein GST-CgENDO1 against nucleic acids with different sources and types, under different Zn^2+^ ions concentrations and pH conditions. The results showed that the genomic DNA of *C. grandis* ‘Tomentosa’ peel cells and of rice leaves (T65) was digestible by the fusion protein at 37 °C for 1 h, at pH 5.5 and 8.0, and when the concentration of Zn^2+^ ions was more than 500 mM (Figure 2Aa,b,2Ba,b). However, at pH 5.5 and 8.0, the fusion protein had a weak digestion effect on the circular BD plasmid DNA when the concentration of Zn^2+^ was more than 1 M (Figure 2Ac,2Bc). Linearizing the BD plasmid by cutting it with the restriction endonuclease EcoRIallowed it to be digested at pH 5.5 and 8.0, with a Zn^2+^ ions concentration greater than 500 mM (Figure 2Ad,2Bd). However, the efficacy was not as good as the digestion of the genomic DNA of both *C. grandis* ‘Tomentosa’ peel cells and of rice leaves (T65). The DNA could not be degraded without the addition of the target fusion protein or Zn^2+^ ions, and the GST protein could not degrade DNA under any conditions. Digestion with DNaseⅠwithout RNase worked on all of the substrates (Figure 2). In the presence of Zn^2+^ ions, at pH 5.5 and 8.0, the degradation of DNA mainly occurred through the fusion protein GST-CgENDO1.

### 3.2. Molecular and Cytological Characteristics of Zn^2+^-Dependent Nuclease CgENDO1 in the Secretory Cavity Cells PCD in C. grandis ‘Tomentosa’ Fruits

We observed changes in the cell structure during the development of the secretory cavity of *C. grandis* ‘Tomentosa’ fruits under a light microscope. In the early initial cell stage, the initial cell group was comprised of 7–10 cells in the exocarp of the fruit. These cells vary in size, were closely arranged, and divided frequently and clearly (Figure 3Aa, arrow). With the enlargement and differentiation of the cell, the typical structure of the secretory cavity consists of a globular part (Figure 3Ab, arrow). In addition, a conical cap part (Figure 3Ab, arrowhead) was formed as the development of the secretory cavity enters the middle initial cell stage (Figure 3Ab). The conical cap part was near the outer epidermis, with narrow and small cells (Figure 3Ab, arrowhead). Several cells in the center of the globular part were large and polygonal, with a close cell arrangement and a dense cytoplasm (Figure 3Ab, arrow). As the secretory cavity continues to develop, the structural differentiation of the globular part and the conical cap part became more obvious, but there was no visible change in the structure of cells. At this time, the development of the secretory cavity enters the late initial cell stage (Figure 3Ac, arrow). Then, it entered the lumen-forming stage, where a small lumen forms (Figure 3Ad, arrow) in the adjacent corners of several cells in the center of the globular part in the secretory cavity, accompanied by small vacuoles in the cytoplasm. With the enlargement of the secretory cavity, it entered the lumen-expanding stage. At this time, the epithelial cells around the lumen had significantly large vacuoles, their cell morphology became irregular, a few cells were destroyed, and the lumen further expands to form a cavity surrounded by 20–30 cells. A large amount of volatile oil accumulated in the secretory cavity and gradually filled the lumen (Figure 3Ae–f).

To further observe the tissue specificity of *CgENDO1* expression at the transcriptional level, in situ hybridization was performed on the secretory cavity at various developmental stages of the exocarp of *C. grandis* ‘Tomentosa’ fruits.

The in situ hybridization experimental results showed that the strongest signal was in the late stage of the initial cell. The signal was slightly weakened in the lumen-forming stage of the secretory cell. Similarly, the signals were weak in the early and middle stages of the initial cell. With secretory cavity development, the signal almost disappeared in the lumen-expanding stage to the mature stage (Figure 3B). There was no in situ hybridization signal in the negative control (Appendix A).

To detect the spatiotemporal variation characteristics of Zn^2+^ ions in the process of PCD of secretory cavity cells, we used the silver amplification method for the first time to convert Zn^2+^ ions in cells into black particles precipitated by Ag_2_S to observe the dynamic change in Zn^2+^ ions. At the same time, the subcellular localization of Zn^2+^-dependent nuclease CgENDO1 was identified using immunocytochemical localization methods to determine its spatiotemporal variation characteristics. Our transmission electron microscopy results revealed a correlation between the spatiotemporal variation of Zn^2+^ ions and Zn^2+^-dependent nuclease CgENDO1.

In the early initial cell stage, the nuclei in the center of the cell, the nuclear membrane, and the nucleolus were clear. The nuclear matrix was dense. There were only a few vacuoles (Figure 4a–h), a tiny amount of silver particles scattered in the cell wall, no silver particles in the cytoplasm matrix and vacuoles (Figure 4b,c, arrowhead), and a small amount of anti-CgENDO1-immunogold particles scattered in the nucleus and vacuoles (Figure 4e–h, arrow). In the middle initial cell stage, the structure of the nucleus was still clear, but chromosomal condensation occurred (Figure 4i–p, rhombus). At the same time, the number of silver particles increased, most of which were still distributed in the cell wall near the cell membrane. Small amounts of silver particles were scattered in the nucleus, cytoplasm, and newly formed vesicles (Figure 4j,k, arrowhead). During this period, more immunogold particles appeared in the nucleus and vacuoles. Most of the immunogold particles were distributed in the nuclear matrix and chromatin condensation. A few immunogold particles appeared in the vacuoles and cytoplasm (Figure 4m–p, arrow). In the late initial cell stage, the boundary of the nucleus of the central cells in the global part became unclear, and the shape of the nucleus was irregular. A few areas of the nuclear membrane were blurred or even broken, and chromatin condensation was marginalized (Figure 5a–h, diamond). There were more silver particles distributed within the cell walls near the cell membrane. A few moved due to free diffusion (Figure 5b,c, arrowhead). A few moved inside by endocytosis (Figure 5b,c, triangle). A few were transported into the cell through plasmodesmata (Figure 5b, star). In addition, a few silver particles were found to occur in intracellular vacuoles (Figure 5b,c, arrowhead). Accordingly, anti-CgENDO1 immunogold particles freely diffused into the nucleus, where numerous immunogold particles were concentrated in the nucleus. A large number of immunogold particles accumulated in vacuoles. A small number of immunogold particles were distributed in the cytoplasm (Figure 5e–h, arrow).

In the lumen-forming stage, in epithelial cells surrounding the lumen, the nucleus was almost degraded, and the chromosomes nucleoli disappeared. The nuclear membrane was fuzzy (Figure 5i–p). A large number of silver particles (Figure 5j, arrowhead) and numerous immunogold particles (Figure 5m,n, arrow) accumulated in the residual nucleus area. A few silver particles (Figure 5, arrowhead) and immunogold particles occurred in the cytoplasmic matrix and vacuoles (Figure 5o,p, arrow).

In the lumen-expanding stage, the nuclei had been completely degraded in the innermost epithelial cells surrounding the lumen (Figure 6a–h), and numerous vesicles occurred in the cell. In this stage, no silver particles were found to occur inside or outside of the cell or in the cell wall (Figure 6b,c). Only the degrading vacuoles and plastids contained a very small amount of immunogold particles (Figure 6e–h, arrow).

To verify the specific distribution of silver particles in secretory cavity cells, we also observed the nonsecretory cavity cells of fruit in the same period and found that only a small amount of silver particles were scattered in the cell wall in the common parenchyma cells, and no silver particles were distributed inside the cytoplasm (Appendix A, arrowhead).

Moreover, to identify the pattern of CgENDO1 protein expression and its tissue specificity, we observed the distribution of anti-CgENDO1-immunogold particles in nonsecretory cavity cells, and found no anti-CgENDO1-immunegold particles (Appendix A). At the same time, immunogold particles were not observed to occur in secretory cavity cells in the control experiment without the *CgENDO1* antibody (Appendix A).

## 4. Discussion

### 4.1. Zn^2+^-Dependent Nuclease *CgENDO1* Is Involved in the Nuclear DNA Degradation Process of Secretory Cavity Cell PCD

Two kinds of divalent cation-dependent nucleases that can degrade dsDNA in plants are Ca^2+^- and Zn^2+^-dependent nucleases. From the perspective of their sequence, most of the Zn^2+^-dependent nucleases are S1/P1-like nucleases and can degrade DNA and RNA in an environment with a pH of 5.5 [15]. The S1/P1-like nuclease in plants contains a conserved sequence with nine amino acid residues, which will bind to three Zn^2+^ ions residues during the enzyme catalytic activity, which is the same position as in S1/P1 nucleases, which are dependent on Zn^2+^ ions to induce the enzyme catalytic activity in a low pH environment [21,45]. The amino acid sequences of CgENDO1, S1, P1, ZEN1, BEN1, and AtENDOs are highly similar, and the nine conserved amino acid residue sequences are completely consistent (Appendix A). Moreover, NCBI prediction results show that CgENDO1 is a S1/P1-like nuclease (Figure 1Bc). The functional domain of CgENDO1 is highly conserved.

The molecular weight of Zn^2+^-dependent nucleases is approximately 33-44 kDa. Zn^2+^ ions can stabilize the enzyme activity and revive enzymes inactivated by EDTA. Zn^2+^-dependent nucleases have the characteristic of 3′-nucleotide enzyme activity, which can notch and linearize double-stranded superhelical DNA, but very few can break double-stranded DNA into small fragments [15]. AtENDOs can hydrolyze nuclear DNA, and the catalytic activities of different members of the same family are different [23]. Among them, AtENDO3 is the only nuclease with similar characteristics as S1/P1 nucleases and mung bean nuclease in fungi, and it has a nuclease activity dependent on Zn^2+^ ions at an acidic pH [21]. AtENDO1 (BFN1) has a Ca^2+^-dependent enzyme activity under neutral conditions and can digest ssDNA in the presence of Ca^2+^ and Mn^2+^ ions, while digestion of dsDNA only requires Mn^2+^ ions [21]. Meanwhile, it can also digest RNA, dsDNA, and ssDNA with Zn^2+^ ions at pH 5.5 and 8.0 [22]. AtENDO2 can digest ssDNA with Mn^2+^ in neutral environments, but at the same time, it can digest dsDNA and ssDNA with Ca^2+^ or Zn^2+^ ions in acidic environments. AtENDO4 can be activated by Mn^2+^ and Ca^2+^ ions at neutral pH. AtENDO5 is a Zn^2+^-dependent nuclease under neutral conditions, and its catalytic efficiency decreases under acidic conditions [21]. Zn^2+^-dependent nucleases play an important role in plant nuclear DNA degradation, for example, ZEN1 can degrade nuclear DNA in acidic environments [13,34,46]. Zn^2+^-dependent nuclease in germinating wheat grains degrades the nuclear DNA of shield and endosperm cells in acidic environments [47]. BEN1 in barley depends on Zn^2+^ ions to break nuclear DNA in endosperm aleurone cells and pollen microgametes of barley [34,35].

Our study found that the molecular weight of CgENDO1 in *C. grandis* ‘Tomentosa’ was 34.45 kDa, and its recombinant protein had an enzyme activity and could degrade the linear double-stranded DNA of *C. grandis* ‘Tomentosa’, T65 genomes, and linear plasmid DNA at both acidic and neutral pH levels with a certain concentration of Zn^2+^ ions, but only had a weak digestion effect on the circle plasmid DNA (Figure 2). This indicates that CgENDO1 prefers to cleave the 5′-phosphomononucleotide and 5′-phosphooligonucleotide end-products [21], and CgENDO1 should be a Zn^2+^-dependent nuclease. Moreover, it was found that CgENDO1 could degrade DNA at pH 5.5 and 8.0, but it requires more Zn^2+^ ions to circle the plasmid DNA (Figure 2). Our results further support the idea that the environment and substrates of different Zn^2+^-dependent nuclease interactions are not identical [48,49]. Therefore, we suggest that CgENDO1 is a Zn^2+^-dependent nuclease capable of degrading linear single-stranded and double-stranded DNA with Zn^2+^ ions in acidic and neutral environments.

The secretory cavity in *Citrus* fruits is formed schizolysigenously, and PCD is involved in the rupture of secretory cavity cells. Cells during PCD have typical chromatin condensation, DNA fragmentation, nuclear degradation, and other important characteristics [12,14,39,40,41,50]. In particular, Ca^2+^ ions and a Ca^2+^-dependent nuclease are involved in the degradation of nuclear DNA during the PCD of secretory cavity cells in *C. grandis* ‘Tomentosa’ fruits [14]. In this study, we found that *CgENDO1* is expressed in *C. grandis* ‘Tomentosa’ fruit secretory cells PCD, especially in the late initial cell stage and the lumen-forming stage (Figure 3B). In addition, the expression patterns of *CgENDO1* show high expression levels in the late initial cell stage and the lumen-forming stage (Figure 1).

Cytological observation showed that the late initial cell stage and the lumen-forming stage were a stage of rapid degradation of the nucleus, which was mainly characterized by the degradation and disappearance of the nucleolus and nuclear membrane, leaving only the residual nuclear region in the cell (Figure 5). In conclusion, the series of changes in nuclear morphology may be related to the high expression level of CgENDO1. Furthermore, the immunocytochemical localization of CgENDO1 showed that there was a large amount of CgENDO1 expression in the nucleus of the secretory cavity cells in the late initial cell stage and the lumen-forming stage. At the onset of the late initial cell stage, a large amount of Zn^2+^ ions was transported from the cell walls to the cytoplasm. Until the lumen-forming stage, a large amount of Zn^2+^ ions were accumulated in the residual nucleus area and the amount of Zn^2+^ ions reached a peak value (Figure 5a–c,i–k). This is the same pattern as the changes in CgENDO1 during secretory cavity cells development. From the middle initial cell stage on, CgENDO1 appeared in the nucleus and vacuole, and began to constantly freely diffuse into the nucleus. As PCD progressed, the amount of CgENDO1 continuously increased in the nucleus until the lumen-forming stage. However, CgENDO1 was sharply reduced in the vacuole of the lumen-forming stage. At this time, a large amount of CgENDO1 was aggregated in the residual nucleus region of the secretory cavity cells (Figure 5d–h,l–p), and should be related to DNA fragment degradation by the enrichment of Zn^2+^ ions involved in the activation of CgENDO1 (Figure 2). Finally, with the complete degradation of the nucleus, both Zn^2+^ ions and CgENDO1 completely disappeared (Figure 6). Therefore, according to the spatiotemporal variation characteristics of Zn^2+^ ions and CgENDO1 as well as the literature analysis, we believe that the Zn^2+^-dependent nuclease CgENDO1, through the activation of Zn^2+^ ions, may be involved in nuclear DNA degradation during secretory cell PCD in *C. grandis* ‘Tomentosa’ fruits, especially participating in residual DNA clearance after the degradation of the nucleolus.

In young and mature plant cells, the pH of the nucleus and cytoplasm is neutral (approximately 7.5), and the concentrations of Ca^2+^ and Zn^2+^ ions are low, but the pH of the plastid and vacuole lumen is approximately 5.5 [51]. Vacuoles are considered storage sites of metal ions, and the Zn^2+^ ions concentration in mature cell vacuoles exceeds the cytoplasm. Similarly, in a normal cell, the Ca^2+^ ions concentration outside the cell is higher than the cytoplasm. However, by coercion or when the cell undergoes some physiological processes, the cell environment will change. For example, in the late stage of PCD, the vacuole membrane is broken to release its contents, which results in an acidic cytoplasm and increases in the number of metal ions [52].

In this study, it was also found that in the early stage of cell PCD, before the need for Zn^2+^ ions to activate CgENDO1, the Zn^2+^ ions were freely dispersed extracellularly (Figure 4a–c), and only a small amount of Zn^2+^ ions was freely dispersed extracellularly in normal nonsecretory cavity cells without PCD (Appendix A). From the middle to late initial cell stage, a large amount of Zn^2+^ ions entered the cell through diffusion, endocytosis, and plasmodesmata transportation (Figure 4i–k and Figure 5a–c), and finally, a large amount of Zn^2+^ ions accumulates in the nuclear region during the lumen-forming stage (Figure 5i–k). After the nucleus is completely degraded in the cells, the Zn^2+^ ions also disappeared after the nuclear rupture (Figure 6a–c). During this process, the Zn^2+^ ions were rapidly transported from the extracellular to the intracellular nuclei during the phase of nuclear degradation, which is consistent with the studies of Martin et al. and Hara-Nishimura and Hatsugai [51,52]. However, our cytochemical results did not show the extensive storage of Zn^2+^ ions in vacuoles or the process of Zn^2+^ ions that are released from the vacuoles. It is speculated that Zn^2+^ ions may exhibit different cytological distribution characteristics during PCD in different types of plant cells.

In conclusion, we believe that the Zn^2+^-dependent nuclease CgENDO1 is involved in the degradation of the nucleus in PCD cells during the development of the secretory cavity in *Citrus grandis* ‘Tomentosa’ fruits, especially during later stages of the degradation of nuclear DNA in PCD cells. It participates in this process by activating CgENDO1 through the abnormal recruitment of Zn^2+^ ions to the nuclear region from the later stage of nuclear degradation to the stage when the nucleus is almost completely degraded.

### 4.2. The Zn^2+^-Dependent Nuclease CgENDO1 and the Ca^2+^-Dependent Nuclease CgCAN May Play a Synergistic Role in the Process of Nuclear DNA Degradation

Four types of nucleases are dependent on divalent cations in plants, among which only Ca^2+^- and Zn^2+^-dependent nucleases are involved in double-stranded DNA degradation [13]. Ca^2+^-dependent nucleases effectively act on double-stranded DNA (dsDNA) under neutral and optimal pH conditions [14]. However, Zn^2+^-dependent nucleases mainly act on single-stranded DNA (ssDNA) and RNA under acidic and optimal pH conditions [15]. Based on the study of Ca^2+^- and Zn^2+^-dependent nucleases in PCD of various plants, Sugiyama et al. proposed a synergistic model of Ca^2+^- and Zn^2+^-nucleases as involved in DNA degradation during PCD [15]. First, at the beginning of PCD, Ca^2+^-dependent nucleases increase in the nucleus, which produce nuclear DNA-limited fragments. In addition, PCD induces Zn^2+^-dependent nucleases synthesis and storage in the cytoplasm or vacuole. Finally, in the later stages of PCD, the membrane system is broken. Nuclear DNA is exposed in the cytoplasm. Zn^2+^-dependent nuclease is released from the plastid or vacuole and activated. The large number of DNA fragments is rapidly and completely degraded by these Zn^2+^-dependent nucleases [15].

Using cytochemistry and immunocytochemistry techniques, Bai et al. found that in the early and middle initial cell stages, the TUNEL signal showed that DNA breaks in the secretory cavity cells appeared in the earliest stage of the initial cells, gradually increased and strengthened, and reached a peak in the middle initial cell stage [14]. At this time, in the secretory cavity cells in the middle initial cell stage, a large amount of Ca^2+^ ions was transferred from the cell wall to the nucleus, and at the same time, a large amount of Ca^2+^-dependent nuclease CgCAN was also transferred to the nucleus. However, we found that only a small amount of Zn^2+^ ions was distributed outside the cytoplasm. In contrast, the abundant Zn^2+^-dependent nuclease CgENDO1 had been present in the nucleus and only a few occurred in vacuoles in the secretory cavity cells in the middle initial cell stage. With the progression of secretory cavity cell PCD, in the late initial cell stage, the TUNEL signal is weakened, and Ca^2+^ ions and Ca^2+^-dependent nuclease CgCAN are rapidly reduced and disappear from the nucleus [14]. These phenomena imply that the double-stranded DNA breakage is almost complete. At this time, the amount of Zn^2+^ ions and CgENDO1 increased significantly. Among them, CgENDO1 mainly accumulated in the nucleus and vacuole, while Zn^2+^ ions mainly accumulated in the margin of the plasma membrane. A few were in the cytoplasm and vacuole, not in the nucleus. In the lumen-forming stage, the nucleus was almost degraded, and the nucleolus and nuclear membrane almost disappeared. At this point, both Zn^2+^ ions and CgENDO1 accumulated in large quantities in the residual region of the nucleus, which in the vacuole were significantly reduced. Less Zn^2+^ ions were in the margin of the plasma membrane (Figure 7).

In conclusion, our results provide direct cytological evidence for the hypothesis of the Ca^2+^- and Zn^2+^-dependent nuclease operating patterns proposed by Sugiyama et al. [15]. Based on the analysis of spatiotemporal variation characteristics of Ca^2+^- and Zn^2+^-dependent nucleases during the process of nuclear degradation in the secretory cavity, we speculate that in PCD cells, Ca^2+^ ions activated Ca^2+^-dependent nucleases to perform double-stranded DNA degradation from the early to middle stage of the secretory cavity development, until the rupture of the nucleus. In the middle to late stage of secretory cavity development, although numerous Zn^2+^-dependent nucleases were concentrated in the nucleus, the structure of the nucleus appeared relatively intact due to only a few Zn^2+^ ions occurring in the nucleus. What is more, a large amount of Zn^2+^-dependent nucleases accumulated in the vacuole. In the lumen-forming stage of secretory cavity cells PCD, when the ruptured nucleus appeared in the residual nuclear region, Zn^2+^ ions were abnormally enriched in the residual nuclear region, which activated Zn^2+^-dependent nucleases in the nucleus and thus, completely degraded the residual nucleic acid fragments. At the moment, Zn^2+^-dependent nucleases in the vacuole sharply decreased, which should be transferred to the residual nuclear region to take part in DNA fragment degradation, since the vacuole membrane was broken. Finally, the nuclear was degraded rapidly.

## Figures and Tables

**Figure 1 cells-10-03222-f001:**
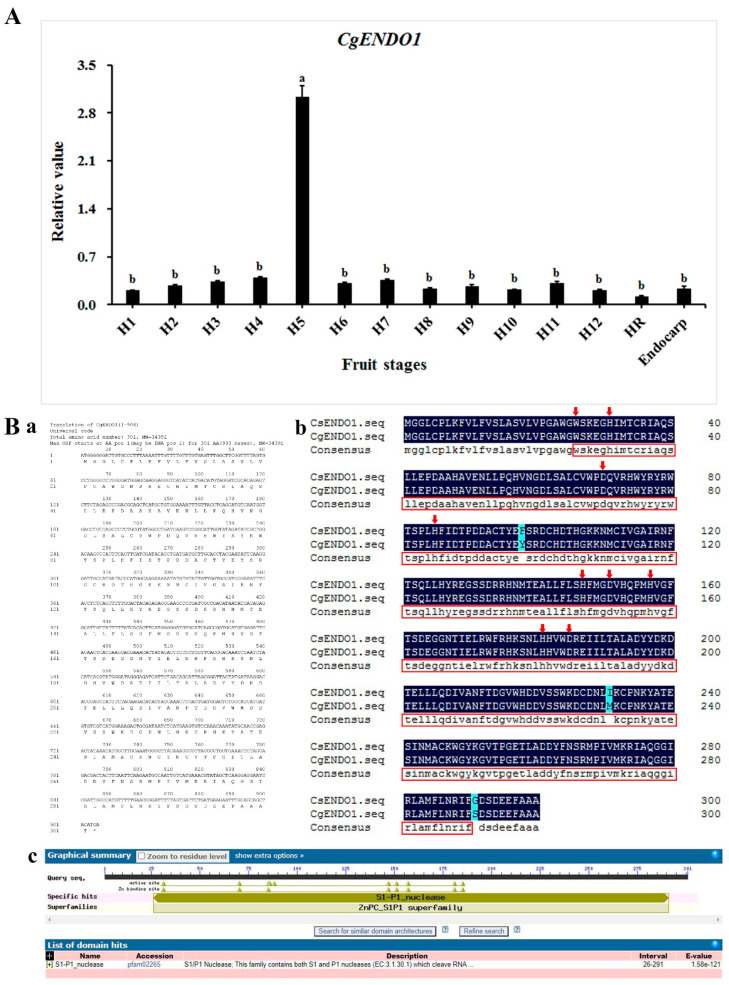
Gene expression and sequence analysis of *CgENDO1*. (**A**) Expression analysis of *CgENDO1* involved in the PCD process of the secretory cavity of *Citrus grandis* ‘Tomentosa’ fruits. CgENDO1 was expressed in each grade of fruit, and the highest expression was found in sample H5 (the sampling statistics showed that H5 was mainly in the late initial cell stage and the lumen-forming stage) (Appendix A). The exocarp of mature fruits and endocarp without a secretory cavity were both expressed at low levels. Different lowercase letters indicate significant differences at *p* ≤ 0.05 (Duncan’s multiple comparison). HR: Ripe fruit. (**B**) Sequence analysis of *CgENDO1* involved in the PCD process of the secretory cavity of *Citrus grandis* ‘Tomentosa’ fruits. (**a**) The cDNA sequence of *CgENDO1* and the amino acid sequence of CgENDO1. The 906 bp cDNA sequence was translated into a protein sequence in the DNAMAN8.0 software. (**b**) The protein sequence of CgENDO1 was compared with CsENDO1 in *Citrus sinensis* (L.) Osbeck. The red rectangle indicates the functional domain of S1/P1 nuclease predicted by InterProScan, and the red arrow indicates that nine conserved amino acid residues predicted by InterProScan bind three Zn^2+^ ions when the S1/P1 nuclease acts. (**c**) NCBI protein BLAST prediction of the CgENDO1 domain and active sites.

**Figure 2 cells-10-03222-f002:**
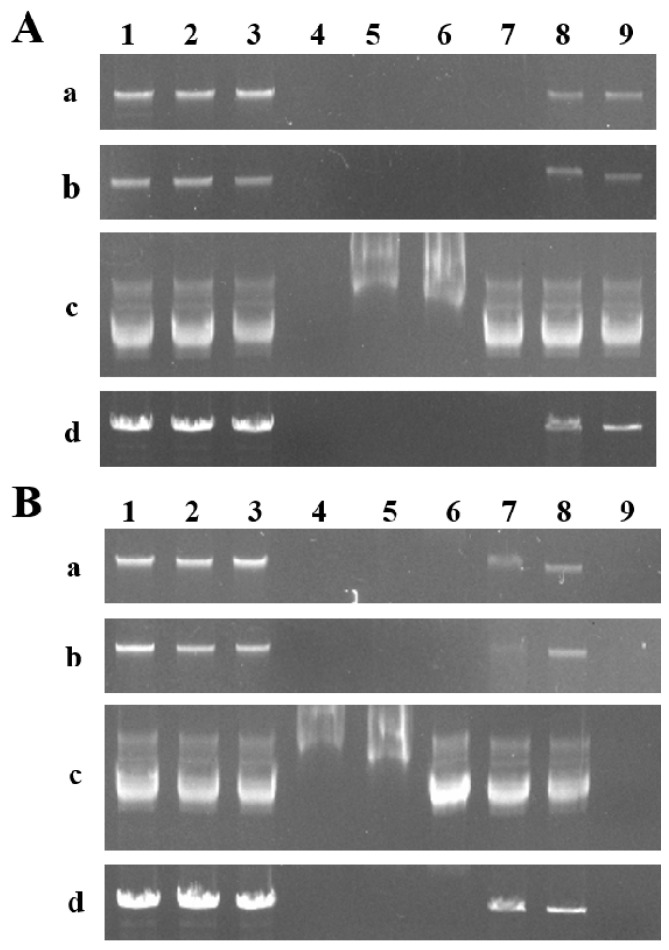
DNase activity analysis of the purified protein GST-CgENDO1 expressed in *Escherichia coli*. (**A**) The fusion protein CgENDO1 digested DNA with different zinc ions concentrations in vitro at pH 5.5. Lanes 1–4: Control group (added protein/Zn^2+^ concentration). Lane 1: Substrate control (-/0); Lane 2: Negative control (0/1 M Zn^2+^); Lane 3: GST protein negative control (GST/1 M Zn^2+^); Lane 4: Positive control; Lanes 5–9: Experimental group, the same amount of purified GST-CgENDO1 protein was added, and the concentrations of Zn^2+^ ions were 2 M, 1 M, 500 mM, 5 mM, and 0. (**a**) *Citrus grandis* ‘Tomentosa’ DNA as the substrate, (**b**) T65 DNA as the substrate, (**c**) circular BD plasmid DNA as the substrate, and (**d**) linear BD plasmid DNA as the substrate. (**B**) The fusion protein CgENDO1 digested DNA with different zinc ion concentrations in vitro at pH 8.0. Lanes 1–3 and Lane 9: Control group (added protein/Zn^2+^ concentration). Lane 1: Substrate control (-/0); Lane 2: Negative control (0/1 M Zn^2+^); Lane 3: GST protein negative control (GST/1 M Zn^2+^); Lane 9: Positive control; Lanes 4–8: Experimental group, the same amount of purified GST-CgENDO1 protein was added, and the concentrations of Zn^2+^ ions were 2 M, 1 M, 500 mM, 5 mM, and 0. (**a**) *Citrus grandis* ‘Tomentosa’ DNA as the substrate, (**b**) T65 DNA as the substrate, (**c**) circular BD plasmid DNA as the substrate, and (**d**) linear BD plasmid DNA as the substrate.

**Figure 3 cells-10-03222-f003:**
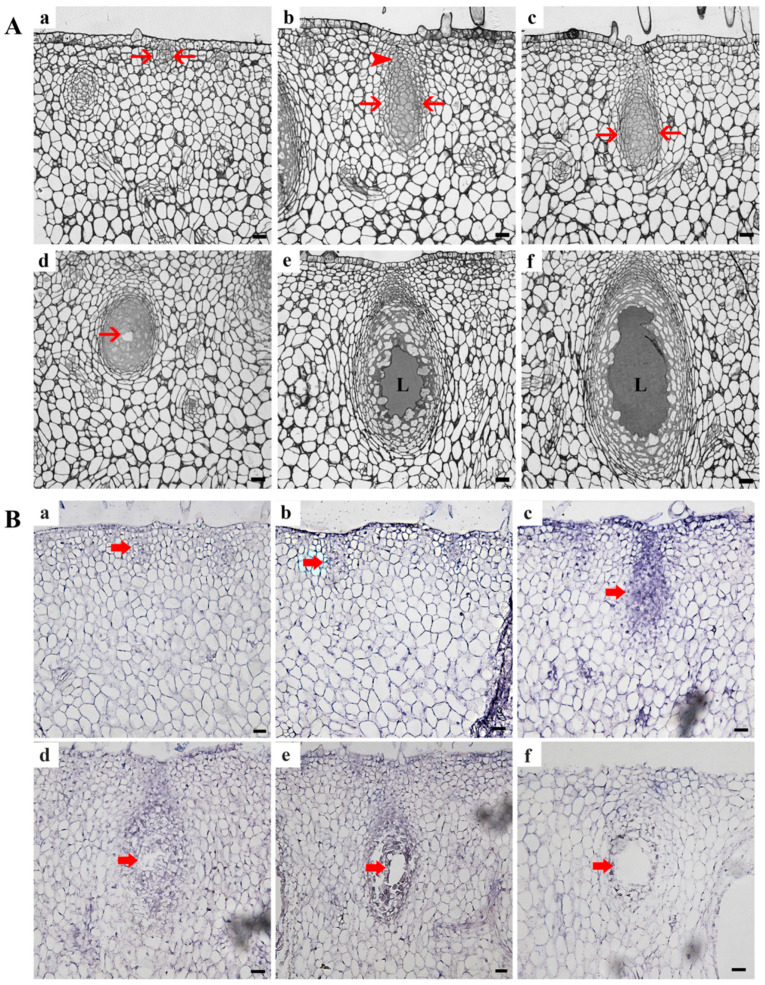
Microscopic results and in situ hybridization analysis of the secretory cavity at different developmental stages of *Citrus grandis* ‘Tomentosa’ fruits. (**A**) Microscopic structure of the secretory cavity in different developmental stages in the exocarp of *Citrus grandis* ‘Tomentosa’ fruits. The initial cell stage of the secretory cavity in (**a**–**c**). (**a**) The early initial cell stage (arrow). (**b**) The middle initial cell stage. A globular part (arrow) and a conical cap part (arrowhead). The cytoplasm of the center cell in the globular part is dense. (**c**) The late initial cell stage. Cytoplasm thinning in the central cell of the globular part (arrow). (**d**) The lumen-forming stage. The arrow represents the newly formed lumen. (**e**,**f**) The lumen-expanding stage. The arrow shows the secretory cavity. (**B**) In situ hybridization analysis of CgENDO1 during the development of the secretory cavity. (**a**–**f**) In situ hybridization signals in secretory cavity cells at different developmental stages. The signals are strongest in the late initial cell stage (**c**), weakened in the lumen-forming stage (**d**) and the lumen-expanding stage (**e**,**f**), and weak in the early initial cell stage (**a**) and the middle initial cell stage (**b**). L: Lumen. Bars = 20 µm.

**Figure 4 cells-10-03222-f004:**
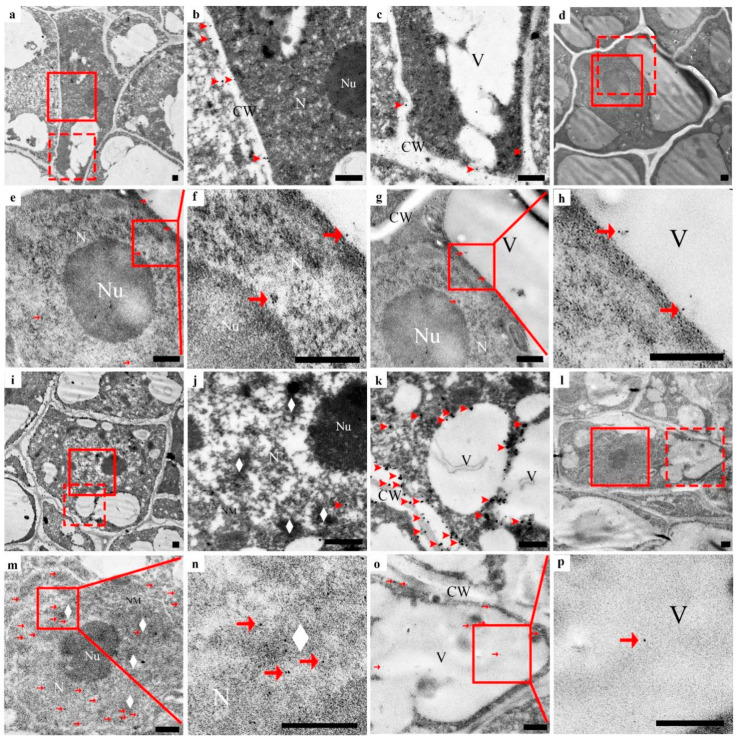
Zn^2+^ ions subcellular localization (**a**–**c**, **i**–**k**) and CgENDO1 immunocytochemistry (**d**–**h**, **l**–**p**) during secretory cavity development of *Citrus grandis* ‘Tomentosa’ fruits. (**a**) Secretory cavity cells in the early initial cell stage. (**b**) Shows the solid line rectangle in (**a**) with a normal nuclear shape and dense nucleoplasm, and (**c**) shows the dotted line rectangle in (**a**) with a small number of silver particles only in the cell wall (arrowhead). (**d**) Shows the secretory cavity cells in the early initial cell stage, (**e**) shows the solid line rectangle of (**d**). (**f**) Shows the solid line rectangle of (**e**), the nucleoli is full, the nucleoplasm is dense, and a very small amount of immunogold particles are in the nucleus (arrow), and (**g**) shows the dotted line rectangle of (**d**). (**h**) Shows the solid line rectangle of (**g**), and a very small amount of immunogold particles are in the vacuole (arrow). (**i**) Central cells of the globulus of the secretory cavity in the middle initial cell stage. (**j**) Shows the solid rectangle in (**i**), the structure of the nucleoli is clear, the chromatin condenses (diamond), and a handful of silver particles are present in the nucleus (arrowhead). (**k**) Shows the dotted rectangle in (**i**), and the double nuclear membrane is clear. Most of the silver particles are in the cell wall, while a few are in vacuole and the cytoplasmic matrix (arrowhead). (**l**) Shows central cells of the globulus of the secretory cavity in the middle initial cell stage. (**m**) Shows the solid line rectangle in (**l**). (**n**) Shows the solid line rectangle of (**m**). The structure of the nucleoli is clear, there is a clear double nuclear membrane and the chromatin condenses (diamond). A more scattered distribution of immunogold particles into the nucleus, and the condensed chromatin has immunogold particles (arrow), (**o**) shows the dotted rectangle in (**l**). (**p**) Shows the solid line rectangle of (**o**), a few of the immunogold particles diffused in the vacuole and the cytoplasmic matrix (arrow). CW: Cell wall; N: Nucleus; NM: Nuclear membrane; Nu: Nucleolus; V: Vacuole. Bars = 500 nm.

**Figure 5 cells-10-03222-f005:**
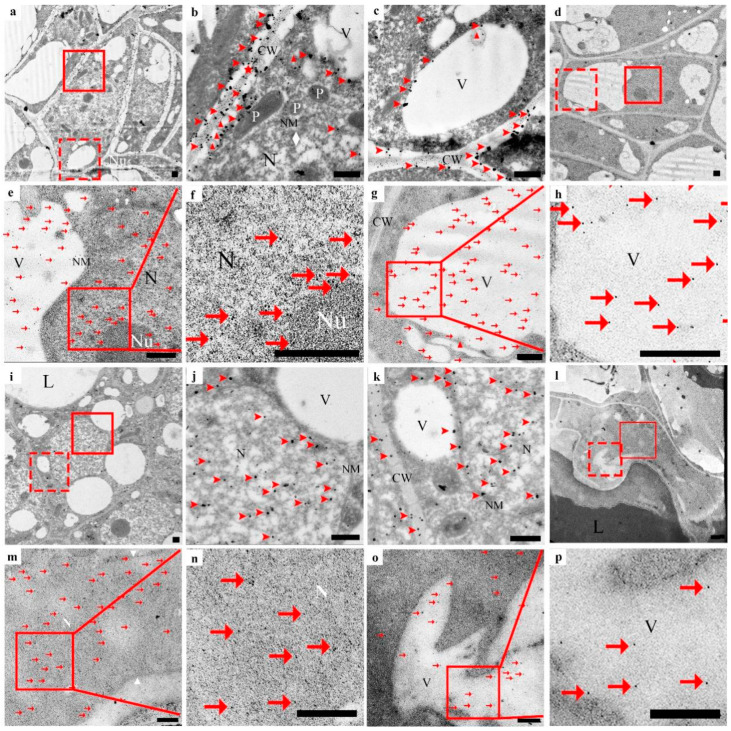
Zn^2+^ ions subcellular localization (**a**–**c**,**i**–**k**) and CgENDO1 immunocytochemistry (**d**–**h**,**l**–**p**) during secretory cavity development of *Citrus grandis* ‘Tomentosa’ fruits. (**a**) Central cells of the globulus of the secretory cavity in the late initial cell stage. (**b**) Shows the solid line rectangle in (**a**). (**c**) Shows the dotted line rectangle in (**a**). The nuclear membrane is partly decomposed, the condensed chromatin has increased (diamond), silver particles by diffusion are entering the cells (arrowhead), and also by phagocytosis (triangle) and plasmodesmata transport (stars) into the vacuole and nuclei (arrowhead). (**d**) Central cells of the globulus of the secretory cavity in the late initial cell stage. (**e**) Shows the solid line rectangle of (**d**). (**f**) Shows the solid line rectangle of (**e**). Degradation of the nuclear membrane decomposition, condensed chromatin (diamond), and a large number of immunogold particles are concentrated in the nucleus, including the nucleoli (arrow). (**g**) Shows the dotted line rectangle of (**d**). (**h**) Shows the solid line rectangle of (**g**), and numerous immunogold particles accumulate in the vacuole. A few are in the cytoplasmic matrix (arrow). (**i**) Shows the innermost secretory cavity cells around the lumen in the lumen-forming stage. (**j**) Shows the solid line rectangle in (**i**), silver particles concentrated in the nucleus residue (arrowhead), and (**k**) shows the dotted rectangle in (**i**). The membrane of the vacuole is degraded. There are silver particles in the cell wall, cytoplasm matrix, and vacuole (arrowhead). (**l**) Shows the innermost secretory cavity cells around the lumen in the lumen-forming stage. (**m**) Shows the solid lines rectangle in (**l**). (**n**) Shows the solid line rectangle of (**m**). Triangles mean the residual nucleus area, where there are numerous immunogold particles (arrow). (**o**) Shows the dotted rectangle in (**l**). (**p**) Shows the solid line rectangle of (**o**), the vacuole is irregular, and less gold particles are distributed in the vacuole and cytoplasmic matrix (arrow). CW: Cell wall; L: Lumen; N: Nucleus; NM: Nuclear membrane; Nu: Nucleolus; P: Plastid; V: Vacuole. Bars = 500 nm.

**Figure 6 cells-10-03222-f006:**
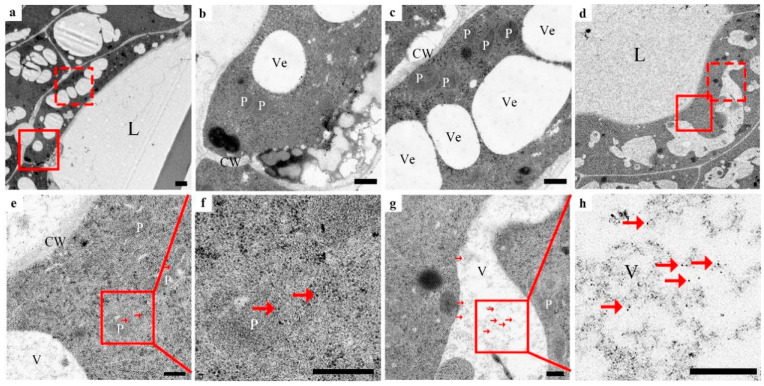
Zn^2+^ ions subcellular localization (**a**–**c**) and CgENDO1 immunocytochemistry (**d**–**h**) during secretory cavity development of *Citrus grandis* ‘Tomentosa’ fruits. (**a**) Shows the innermost secretory cavity cells around the lumen in the lumen-expanding stage. (**b**) Shows the solid line rectangle in (**a**). (**c**) Shows the dotted line rectangle in (**a**). There were no silver granules in the cell wall, vesicle, and cytoplasmic matrix. (**d**) Shows the innermost secretory cavity cells around the lumen in the lumen-expanding stage. (**e**) Shows the solid line rectangle of (**d**); (**f**) shows the solid line rectangle of (**e**); a few gold particles remain in the plastids with unclear structures (arrow). (**g**) Shows the dotted line rectangle of (**d**); (**h**) shows the solid line rectangle of (**g**); there are very few gold particles in the vacuoles that were destroyed (arrow). CW: Cell wall; L: Lumen; N: Nucleus; NM: Nuclear membrane; Nu: Nucleolus; P: Plastid; V: Vacuole; Ve: Vesicle. Bars = 500 nm.

**Figure 7 cells-10-03222-f007:**
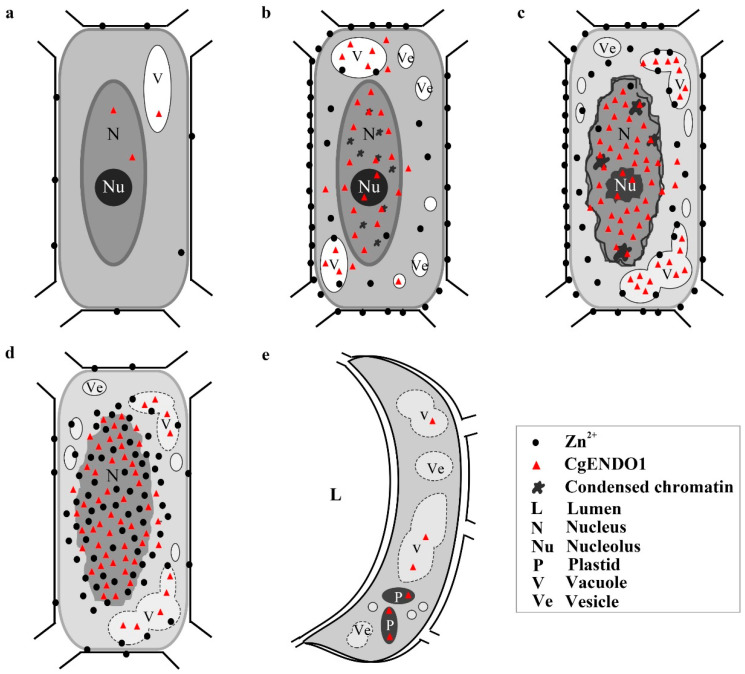
The pattern of Zn^2+^- ions and Zn^2+^-dependent nuclease 1 (CgENDO1) involved in PCD in the development of the secretory cavity of *Citrus grandis* ‘Tomentosa’ fruits. In the early initial cell stage (**a**), very little Zn^2+^ ions and CgENDO1 are in the cell. In the middle initial cell stage (**b**), a large amount of Zn^2+^ ions is in the cell wall and a large amount of CgENDO1 is in the nucleus and vacuole. In the late initial cell stage (**c**), the Zn^2+^ ions are transferred into the cytoplasmic matrix and vacuole, while CgENDO1 is heavily concentrated in the nucleus and vacuole. In the lumen-forming stage (**d**), Zn^2+^ ions are transferred into the nucleus area, and CgENDO1 is concentrated in the nucleus at the same time, while less CgENDO1 is in the vacuole. In the lumen-expanding stage (**e**), the Zn^2+^ ions completely disappear, with only a small amount of CgENDO1 in the vacuole.

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
