# Peer review of "Zn2+-Dependent Nuclease Is Involved in Nuclear Degradation during the Programmed Cell Death of Secretory Cavity Formation in Citrus grandis ‘Tomentosa’ Fruits"

_cells, 2021, doi:10.3390/cells10113222_

Round 1

Reviewer 1 Report

Dear Editor,

the article “Zn2+-dependent nuclease is involved in nuclear degradation 2
during the programmed cell death of secretory cavity formation 3
in Citrus grandisTomentosa’ fruits ” is an interesting description of PCD events necessary to form secretory cavities in Citrus grandis. The fact that PCD is involved in the process is not surprising, since canals formation in plants is often due to programmed cell death (in other cases to wall dissolution as remembered by the authors). However, here the authors also link the effect of a Zn2+ dependent nuclease to the PCD that is a novel result.

The article is relevant for Cells in my opinion and I suggest publication after a revision. The English text is in general good but can be improved in some points by changing the choice of words. For instance: “whether Zn2+-dependent nuclease plays a role in the PCD of secretory cells has not been reported “ or “Programmed cell death (PCD) refers to suicide“: better to rephrase.

In the introduction and in the description of the phenomenon it may be useful to cite the formation of apoptotic bodies in animals but not in plants, where cell shrinkage occurs (as for instance in Papini A. (2018) The investigation of morphological features of autophagy during plant Programmed Cell Death (PCD). In De Gara L, Locato V (Eds.) “Plant Programmed Cell Death: Methods and Protocols”. Springer Verlag). The text may be useful also for the use of the term programmed cell death instead of apoptosis in plants.

The value of the article is somewhat limited by the fact that the results are based on samples derived from a single individual, but the amount of data produced is, however, huge and hence the results may be considered not yet definitive but relevant.

In Fig. 5F (that is important) I could not see well the particles accumulated in the vacuole. Maybe too many red arrows and probably better to increase the magnification. Possibly dividing figure 5 in two, with images at higher magnification. The same for figure 6. Maybe in the final version, the images are of better quality, but currently, the described features are not clearly visible.

I appreciated Fig. 7, which is very explicative.

Pg 2 line 50: “caspase-like proteases “ are metacaspase cited the line before

pg 3 line 99 “In PCD cells, nuclear degradation “ rephrase

pg 5: rephrase “Total RNA was extracted by the Column Plant RNAout 2.0 kit

pg 6: rephrase “The sections were cut into 1 μm sections

pg 6: rephrase “We used the silver amplification method of Pihl E.: Heavy metals were transferred to silver sulfide for subcellular localization “

pg 16: improve the sentence “when experiencing some physiological processes in cells, the cell environment will change.

pg 17: “the Zn2+ also disappeared “->”the Zn2+ ions also disappeared “

Author Response

Dear reviewer:

Thank you very much for giving us an opportunity to revise our manuscript. According to the reviewer’s comments, we have carefully revised our manuscript to address the comments. Below are our point-by-point responses to each comment:

Thank you for your review and comments

(1) “whether Zn2+-dependent nuclease plays a role in the PCD of secretory cells has not been reported” rephrase

[Answer]: First, thank you for your comments. As suggested, we rephrased “whether Zn2+-dependent nuclease plays a role in the PCD of secretory cells has not been reported” into “whether Zn2+-dependent nuclease plays a role in the PCD of secretory cells remains poorly understood”. (seeing line 16-17)

(2) “Programmed cell death (PCD) refers to suicide” rephrase

[Answer]: First, thank you for your comments. As suggested, we rephrased “Programmed cell death (PCD) refers to suicide …” into “Programmed cell death (PCD) is an intracellular program for death, which means a cell executes a specific physiological process under the control of its own genes.” (seeing line 32-33)

(3) The text may be useful also for the use of the term programmed cell death instead of apoptosis in plants.

[Answer]: First, thank you for your suggestion. As suggested, we used PCD instead of apoptosis in plants in the text.

(4) In Fig. 5F (that is important) I could not see well the particles accumulated in the vacuole. Maybe too many red arrows and probably better to increase the magnification. Possibly dividing figure 5 in two, with images at higher magnification. The same for figure 6. Maybe in the final version, the images are of better quality, but currently, the described features are not clearly visible.

[Answer]: Thank you very much for your valuable suggestion. As suggested, Fig. 4, Fig. 5 and Fig. 6 were carefully modified. We provided images at higher magnification for immunocytochemical localization so that the immunogold particles could be more clearly visible..

(5) Pg 2 line 50: “caspase-like proteases ” are metacaspase cited the line before

[Answer]: First, thank you for your comments. According to current research, caspases, metacaspases and paracaspases constitute a large family named C14. They are within CD clan cysteine proteases. They all share tertiary structure but exhibit variation in structural topology, substrate specificity and activation mechanisms [1]. On the one hand, in fungi and plants, caspase-like activity during PCD was shown to originate from proteases not belonging to the C14 family [2, 3]. As a consequence, the caspase-like activities reported to be involved in plant and fungal cell death most probably differ from the metacaspases [4-11].

References:

  1. McLuskey, K.; Mottram, J. C. Comparative structural analysis of the caspase family with other clan CD cysteine peptidases. Biochem J. 2015, 466: 219-232. doi: https://doi.org/10.1042/BJ20141324
  2. van Doorn, W. G.; Beers, E. P.; Dangl, J. L.; Franklin-Tong, V. E.; Gallois, P.; Hara-Nishimura, I.; Jones, A. M.; Kawai-Yamada, M.; Lam, E.; Mundy, J.; Mur, L. A.; Petersen, M.; Smertenko, A.; Taliansky, M.; Van Breusegem, F.; Wolpert, T.; Woltering, E.; Zhivotovsky, B.; Bozhkov, P. V. Morphological classification of plant cell deaths. Cell Death Differ. 2011, 18: 1241-1246, doi: https://doi.org/10.1038/cdd.2011.36
  3. Hatsugai, N.; Yamada, K.; Goto-Yamada, S.; Hara-Nishimura, I. Vacuolar processing enzyme in plant programmed cell death. Front Plant Sci. 2015, 6: 234, doi: https://doi.org/10.3389/fpls.2015.00234
  4. Vercammen, D.; van de Cotte, B.; De Jaeger, G.; Eeckhout, D.; Casteels, P.; Vandepoele, K.; Vandenberghe, I.; Van Beeumen, J.; Inzé, D.; Van Breusegem, F. Type II metacaspases Atmc4 and Atmc9 of Arabidopsis thaliana cleave substrates after arginine and lysine. J Biol Chem. 2004, 279: 45329-45336, doi: https://doi.org/10.1074/jbc.M406329200
  5. Vercammen, D.; Belenghi, B.; van de Cotte, B.; Beunens, T.; Gavigan, J. A.; De Rycke, R.; Brackenier, A.; Inzé, D.; Harris, J. L.; Van Breusegem, F. Serpin1 of Arabidopsis thaliana is a suicide inhibitor for metacaspase 9. J Mol Biol. 2006, 364: 625-636, doi: https://doi.org/10.1016/j.jmb.2006.09.010
  6. Bozhkov, P. V.; Suarez, M. F.; Filonova, L. H.; Daniel, G.; Zamyatnin, A. A.; Jr, Rodriguez-Nieto, S.; Zhivotovsky, B.; Smertenko, A. Cysteine protease mcII-Pa executes programmed cell death during plant embryogenesis. Proc Natl Acad Sci U S A. 2005, 102: 14463-14468, doi: https://doi.org/10.1073/pnas.0506948102
  7. González, I. J.; Desponds, C.; Schaff, C.; Mottram, J. C.; Fasel, N. Leishmania major metacaspase can replace yeast metacaspase in programmed cell death and has arginine-specific cysteine peptidase activity. Int J Parasitol. 2007, 37: 161-172, doi: https://doi.org/10.1016/j.ijpara.2006.10.004
  8. Watanabe, N.; Lam, E. Two Arabidopsis metacaspases AtMCP1b and AtMCP2b are arginine/lysine-specific cysteine proteases and activate apoptosis-like cell death in yeast. J Biol Chem. 2005, 280: 14691-14699, doi: https://doi.org/10.1074/jbc.M413527200
  9. Coffeen, W. C.; Wolpert, T. J. Purification and characterization of serine proteases that exhibit caspase-like activity and are associated with programmed cell death in Avena sativa. Plant Cell. 2004, 16: 857-873, doi: https://doi.org/10.1105/tpc.017947
  10. Hara-Nishimura, I.; Hatsugai, N.; Nakaune, S.; Kuroyanagi, M.; Nishimura, M. Vacuolar processing enzyme: an executor of plant cell death. Curr Opin Plant Biol. 2005, 8: 404-408. doi: https://doi.org/10.1016/j.pbi.2005.05.016
  11. Hatsugai, N.; Kuroyanagi, M.; Nishimura, M.; Hara-Nishimura, I. A cellular suicide strategy of plants: vacuole-mediated cell death. Apoptosis. 2006, 11: 905-911, doi: https://doi.org/10.1007/s10495-006-6601-1

(6) pg 3 line 99 “In PCD cells, nuclear degradation ” rephrase

[Answer]: Thanks for your comments. As suggested, we rephrased “In PCD cells, nuclear degradation …” into “Multiple nucleases were required for nuclear degradation in PCD cells”. (seeing line 99)

(7) pg 5: rephrase “Total RNA was extracted by the Column Plant RNAout 2.0 kit”

[Answer]: Thanks for your comments. As suggested, we rephrased “Total RNA was extracted by the Column Plant RNAout 2.0 kit” into “Total RNA was extracted using the Column Plant RNAout 2.0 kit (TIANDZ)”. (seeing line 129)

(8) pg 6: rephrase “The sections were cut into 1 μm sections"

[Answer]: Thanks for your comments. As suggested, we rephrased “The sections were cut into 1 μm sections” into “The samples were cut into 1 μm thick sections using a Leica RM2155 thin slicer (Leica, Germany)”. (seeing line 170-171)

(9) pg 6: rephrase “We used the silver amplification method of Pihl E.: Heavy metals were transferred to silver sulfide for subcellular localization"

[Answer]: Thanks for your comments. As suggested, we rephrased “We used the silver amplification method of Pihl E.: Heavy metals were transferred to silver sulfide for subcellular localization” into “We used the silver amplification method described by Pihl E. that heavy metals were transformed into silver sulfide for subcellular localization[44]”. (seeing line 196-197)

(10) pg 16: improve the sentence “when experiencing some physiological processes in cells, the cell environment will change.”

[Answer]: Thanks for your comments. As suggested, we rephrased “when experiencing some physiological processes in cells, the cell environment will change.” into “when the cell undergoes some physiological processes, the cell environment will change.” (seeing line 521-522)

(11) pg 17: “the Zn2+ also disappeared“->”the Zn2+ ions also disappeared “

[Answer]: Thanks for your comments. As suggested, we modified all ions in the full text.

Reviewer 2 Report

It is an interesting and quite well-written manuscript. There is no doubt that experiments were well planned and executed, and the results are well described and discussed. In my opinion, the high value of the manuscript is a merge of many different techniques giving results on molecular and cytologic levels. Generally, I have no objections, and I recommend the publication of the manuscript. However, I have some minor editorial recommendations.
•    All keywords must be changed because keywords should not be the same as words already used in a title.
•    Figure 1 must be moved from Materials and methods to the Results section.
•    Table S1 is repeated. It is presented in the Materials and methods (section 2.2) and it is repeated in the supplementary materials. It must be removed from one of these places. Removing from section 2.2 will be better.
•    Figure 2 also should be moved from Materials and methods to the Results, and additionally, it can be a bit smaller.
•    I recommend changing the end of the Discussion. First, ending with reference looks not good, and secondly, the authors should consider adding the separate section Conclusions.

Author Response

Dear reviewer:

Thank you very much for giving us an opportunity to revise our manuscript. According to the reviewer’s comments, we have carefully revised our manuscript to address the comments. Below are our point-by-point responses to each comment:

Thank you for your review and comments.

(1) All keywords must be changed because keywords should not be the same as words already used in a title.

[Answer]: Thank you very much for your comments. As suggested, we rewrote keywords as follows:

“secretory cavity; PCD; CgENDO1; Zn2+ ions; nuclear DNA degradation.” (seeing line 29)

(2) Figure 1 must be moved from Materials and methods to the Results section.

[Answer]: Thank you very much for your valuable suggestion. As suggested, Figure 1 was moved to Results (section 3.1). (seeing line 262-274)

(3) Table S1 is repeated. It is presented in the Materials and methods (section 2.2) and it is repeated in the supplementary materials. It must be removed from one of these places. Removing from section 2.2 will be better.

[Answer]: Thank you for your valuable suggestion. As suggested, Table S1 in the Materials and methods (section 2.2) was deleted.

(4) Figure 2 also should be moved from Materials and methods to the Results, and additionally, it can be a bit smaller.

[Answer]: Thanks for your comments. As suggested, Figure 2 was moved to Results (section 3.1) and it was smaller. (seeing line 275-288)

(5) I recommend changing the end of the Discussion. First, ending with reference looks not good, and secondly, the authors should consider adding the separate section Conclusions.

[Answer]: Thanks for your comments. As suggested, we have changed the end of the Discussion as follows.

“In conclusion, our results provide direct cytological evidence for the hypothesis about Ca2+- and Zn2+-dependent nuclease operating patterns proposed by Sugiyama et al. [15]. Based on the analysis of spatiotemporal variation characteristics of Ca2+- and Zn2+-dependent nucleases during the process of nuclear degradation in the secretory cavity, we speculate that in PCD cells, Ca2+ ions activated Ca2+-dependent nucleases to perform double-stranded DNA degradation from the early to middle stage of the secretory cavity development until the rupture of the nucleus. In the middle to late stage of secretory cavity development, though numerous Zn2+-dependent nucleases were concentrated in the nucleus, the structure of the nucleus appeared relative intact due to only a few Zn2+ ions occurring in nucleus. What’s more, lots of Zn2+-dependent nucleases accumulated in the vacuole. In the lumen-forming stage of secretory cavity cells PCD, when the ruptured nucleus appeared in the residual nuclear region, Zn2+ ions was abnormally enriched in the residual nuclear region, which activated Zn2+-dependent nucleases in the nucleus and thus completely degraded the residual nucleic acid fragments. At the moment, Zn2+-dependent nucleases in the vacuole sharply decreased, which should be transferred to the residual nuclear region to take part in DNA fragment degradation because the vacuole membrane was broken. At last, the nuclear was degraded rapidly.” (seeing line 591-607)